# Green Tea Extract in the Extender Improved the Post-Thawed Semen Quality and Decreased Amino Acid Mutation of Kacang Buck Sperm

**DOI:** 10.3390/vetsci9080403

**Published:** 2022-08-01

**Authors:** Suherni Susilowati, Imam Mustofa, Wurlina Wurlina, Tatik Hernawati, Yudit Oktanella, Soeharsono Soeharsono, Djoko Agus Purwanto

**Affiliations:** 1Division of Veterinary Reproduction, Faculty of Veterinary Medicine, Universitas Airlangga, Surabaya 60115, Indonesia; suherni-s@fkh.unair.ac.id (S.S.); wurlina@fkh.unair.ac.id (W.W.); tatik-h@fkh.unair.ac.id (T.H.); 2Division of Veterinary Reproduction, Faculty of Veterinary Medicine, Brawijaya University, Malang City 65145, Indonesia; yudito@ub.ac.id; 3Division of Veterinary Anatomy, Faculty of Veterinary Medicine, Universitas Airlangga, Surabaya 60115, Indonesia; soeharsono@fkh.unair.ac.id; 4Division of Pharmaceutical Chemistry, Faculty of Pharmacy, Universitas Airlangga, Surabaya 60115, Indonesia; djoko-a-p@ff.unair.ac.id

**Keywords:** extinction risk, smallholder farmer, poverty reduction, sperm motility, sperm viability

## Abstract

**Simple Summary:**

The Kacang goat (*Capra hircus*) is a livestock species native to Indonesia, and its genetic resources must be protected and preserved to prevent extinction. Frozen semen is the appropriate choice for artificial insemination in Kacang goats. Post-thawed semen can be used more than 20 years after semen collection and can cover a large area of acceptors. However, frozen semen of the Kacang buck is currently unavailable. This was the first study involving the mutation of the amino acids as a quality parameter of post-thawed Kacang buck semen. The results showed that post-thawed Kacang buck semen that was frozen in a 0.10 mg green tea extract per 100 mL skim milk–egg yolk extender and that was equilibrated for one hour resulted in the highest semen quality and lowest amino acids mutation. Glycine, valine, leucine, serine, and asparagine strongly correlate to post-thawed sperm motility. It can be concluded that antioxidants helped improve the quality of post-thawed Kacang buck semen. This finding can be used by the Artificial Insemination Center in Indonesia to produce Kacang buck frozen semen for artificial insemination, which can then be massively applied in the field to increase the population and genetic quality of Kacang goats in time to prevent risk of extinction.

**Abstract:**

This study was the first to combine the addition of antioxidants to a skim milk–egg yolk (SM–EY) extender and different equilibration periods to obtain higher quality post-thawed Kacang buck semen. This study aimed to determine the effects of green tea extract (GTE) on the quality of frozen Kacang goat sperm equilibrated for one and two hours. The pool of Kacang buck ejaculate was equally divided into four portions and was diluted in an SM–EY extender that contained four doses of 0, 0.05, 0.10, and 0.15 mg of GTE/100 mL for T0, T1, T2, and T3 groups, respectively. The aliquots were treated for an equilibration period of 1–2 h before further processing as frozen semen. Post-thawed semen quality was evaluated for sperm quality. The Sanger method was used for DNA sequencing, and the amino acid sequence was read using MEGA v.7.0. The post-thawed semen of the T2 group that was equilibrated for one hour had the highest semen quality. Pre-freezing motility had the highest determination coefficient compared to post-thawed sperm motility. This study is the first to report amino acid mutation due to freeze–thawing. The frequency of amino acid mutations revealed that T2 was the least mutated amino acid. Glycine, valine, leucine, serine, and asparagine strongly correlated to post-thawed sperm motility. It can be concluded that a combination of 0.1 mg GTE/100 mL extender as an antioxidant and one-hour equilibration period resulted in the best post-thawed Kacang buck semen quality.

## 1. Introduction

The Kacang goat (*Capra hircus*) is a critically endangered livestock species native to Indonesia, and its genetic resources must be protected and preserved to prevent the risk of extinction [1]. Smallholder farmers rear Kacang goats as a source of income for poverty reduction. Frozen semen is the appropriate choice for artificial insemination in goats. Post-thawed semen can be used for up to 20 years after semen collection and can cover a large area of acceptors. However, frozen semen of the Kacang buck is currently unavailable. The sperm of Kacang bucks and other goats are sensitive to cryopreservation. The enzyme phospholipase A, secreted from the bulbourethral gland [2], causes it to interact with phospholipids during the freezing of semen, decreasing semen quality [3]. During the freezing process, semen is subjected to cold shock and osmotic and oxidative stresses [4]. Previous studies have found that sperm motility in fresh Kacang goat semen is about 90% [5]. Conversely, after thawing frozen Kacang goat sperm with an antioxidant-free citrate egg yolk enhancer, sperm motility rapidly decreased to 38.50% ± 0.70% [5]. Post-thawed semen motility to be used for artificial insemination (AI) must be more than 40% [6]; thus, Kacang buck semen extended in egg yolk–citrate without antioxidants is not viable for AI. Green tea extract (GTE) contains epigallocatechin-3-gallate (EGCG), which is a powerful antioxidant that has been shown to increase the quality of ram semen after thawing [7], reduce oxidative stress parameters, enhance sperm viability, motility, and the intact plasma membrane, and finally improve the fertility of buffalo sperm [8].

Additionally, the equilibration period for the freezing process of Kacang buck semen has not yet been studied, whereas for other livestock species, there have been several studies on equilibration periods, even though they have been established as freezing protocols. Doležalová et al. [9] reported that an equilibration period of 240 min for bull semen resulted in the best post-thawed sperm motility. Meanwhile, Dwinofanto et al. [10] concluded that Bali bull semen had higher post-thawed quality when equilibrated for 22 h before freezing compared with only four hours.

Mitochondrial molecular markers reflect semen quality. The respiratory chain components needed for oxidation energy metabolism are encoded in the mitochondrial DNA (mtDNA). Male infertility has been associated with mtDNA genetic variants [11]. Mutations in sperm mtDNA are correlated with male fertility [12]. The variation in the NADH dehydrogenase 1 (ND1) of mtDNA genes may cause a decrease in mitochondrial respiratory chain complex activities [13]. Our previous study revealed that GTE increases the quality and reduces DNA mutations of post-thawed Kacang buck sperm [14]. However, the amino acid encoders of the mtDNA mutation in post-thawed livestock animals have not yet been discovered. Therefore, this study aimed to find the effective dose of GTE in the SM–EY extender and the equilibration period to obtain the best quality of Kacang goat semen based on sperm viability, motility, intact plasma membrane (IPM), malondialdehyde (MDA), sperm DNA fragmentation (SDF), sperm capacitated status, and acrosome reaction. In addition, we sought to determine the amino acid that encodes the ND1 mtDNA mutation pattern of post-thawed Kacang buck sperm and its correlation with semen quality parameters.

## 2. Materials and Methods

This research on Kacang buck semen preservation was a multi-year study that started in 2019. The Animal Research Ethics Committee of Airlangga University approved this research procedure, No.520/HRECC.FODM/VII/2019. The commission assessed this study’s proposal based on animal welfare principles. The collection of buck semen was conducted following the World Organization for Animal Health’s Terrestrial Animal Health Act, Protocol of Chapter 4.7 (Cow, Small Ruminant, Pig Semen Collection, and Processing) of [5].

This study was conducted from April 2021 to February 2022 at The Regional Artificial Insemination Center, the Faculty of Veterinary Medicine, Universitas Airlangga, Tanjung village, Kedamean, District of Gresik, East Java, Indonesia. Four heads of Kacang bucks aged 2–3 years, weighing 35–45 kg, were reared in individual stalls, fed approximately 5 kg forage and 0.5 kg concentrate (16–18% crude protein) daily, with always available drinking water. Semen collection was conducted twice a week using an artificial vagina. The ejaculates were evaluated for progressive individual movements first, and if it was more than 70%, it was further processed as frozen semen. A pool of 12 ejaculates was divided equally into four portions for this study (Figure 1).

### 2.1. GTE Preparation and Extender

Dried green tea leaves were obtained from a tea plantation at Nglinggo Barat, Pagerharjo, Kec. Samigaluh, Kulon Progo Regency, Special Region of Yogyakarta, Indonesia. The tea plantation is geographically located between 7°38′42″–7°59′3″ south latitude and 110°1′37″–110°16′26″ east longitude, with an altitude of 600–700 m above sea level. The green tea leaves were extracted using ethanol solvent and the previous study. Dried green tea (*Camellia sinensis* L. Kuntze) leaves of 1.5 kg were ground to a particle size of 0.75 µm, soaked in 96% ethanol, aluminum foil-covered, and left for three days. This solution was filtered with Whatman paper 0.5 microns, and then, the filtrate was evaporated on a rotary evaporator at 45 rpm at 50 °C for 30 min. The extract was evaporated in an acidic chamber until a solid extract was formed; then, it was freeze-dried and stored at −20 °C until further used [15].

Next, 10 g of skim milk powder (Merck 115338) was dissolved in distilled water up to 100 mL, heated at 92 °C to 95 °C for 10 min, and then cooled to 37 °C.

Then, 95 mL of the skim milk solution was added to 5 mL of chicken egg yolk, which was followed by penicillin (1000 IU/mL) and streptomycin (1 mg/mL). Furthermore, the extender was divided into four equal parts, a control group (CG, no GTE) and groups T1, T2, and T3, with 0.05, 0.1, and 0.15 mg GTE added for each 100 mL extender.

### 2.2. Frozen Semen

Each group extender, as mentioned above, was divided into two parts equally. According to the group, the first part was added to fresh semen to obtain a 480 million sperm/mL concentration. The second was added to glycerol to a concentration of 16%. Then, it was slowly added to the first portion of the same amount to reach the semen concentration of 240 million sperm/mL. Next, all groups of extended semen were divided again into two equal parts for subgroup equilibration at 5 °C in the Cold Handling Cabinet (Minitube) for one or two hours, respectively. Post-equilibration of each subgroup was evaluated for the pre-freezing semen quality; then, it was filled into 0.25 mL straws (IMV Technologies, L’Aigle, France) containing 60 million sperm/straw. The filled straws were exposed to liquid nitrogen vapor (2 cm on the surface of liquid nitrogen, −140 °C) for 10 min and stored immediately in liquid nitrogen (−196 °C). The Kacang buck-frozen semen was stored for two weeks before being assessed for post-thawed semen quality. First, thawing was conducted in sterile water at 37 °C for 30 s. Then, sperm viability, progressive motility, IPM, MDA levels, and DNA fragmentation were evaluated according to the previously described methods [15].

### 2.3. Semen Quality Assessment

The sperm viability, progressive motility, percentage of IPM, MDA levels, sperm capacitation and acrosome reaction were measured based on previous reports [16].

#### 2.3.1. Sperm Viability

Dried-smeared semen samples were stained with eosin–nigrosin and then examined under a light microscope (Olympus BX-53, Shinjuku City, Tokyo, Japan) at 400× magnification for 100 sperm. Live sperm is bright, transparent in their heads, and dead sperm is reddish [16].

#### 2.3.2. IPM

A 0.1 mL semen sample was diluted in 1 mL of hypoosmotic solution (1.352 g fructose and 0.735 g sodium citrate 2.H_2_O dissolved in 100 mL distilled water); then, it was incubated at 37 °C for 30 min. The sperm with an intact plasma membrane was a rounded tail, and the damaged sperm plasma membrane was a straightened tail. The IPM of 100 spermatozoa was counted at 400× magnification under a light microscope (Olympus BX-53, Shinjuku City, Tokyo, Japan) [16].

#### 2.3.3. Malondialdehyde Levels

The MDA levels were measured using the thiobarbituric acid (TBA) method. Standards of MDA kit, 8 µg/mL TBA, and 100 µL semen samples were dissolved in distilled water up to 550 µL; then, 100 µL of 20% trichloroacetic acid was added and it was vortexed for 30 s. Then, 250 µL of HCl 1N was added to the mixture and homogenized, after which 100 µL of 1% sodium thiobarbiturate was added, homogenized, and centrifuged at 31× *g* for 10 min. The supernatant was incubated in a water bath at 100 °C for 30 min and then left to room temperature. The kit and sample absorbance was detected using a spectrophotometer at 533 nm wavelength. The MDA levels (ng/mL) were determined based on the extrapolation of absorbance values of the samples on the standard MDA curve [16].

#### 2.3.4. Sperm DNA Fragmentation

The DNA fragmentation was measured using the toluidine blue staining method. A dried–smeared drop of sample in a slide was fixed in 96% ethanol at 4 °C for 30 min, air-dried, and then hydrolyzed in HCl 0.1 N at 4 °C for 5 min. We rinsed the slides three times with distilled water and stained with 0.05% toluidine blue stain for 10 min, washed with distilled water, dehydrated using t-butanol, and cleaned twice with xylol. Sperm with normal DNA were stained blue, whereas DNA fragmented sperm were stained dark blue in sperm heads. The SDF was examined for 100 spermatozoa at 400× magnification under the light microscope (Olympus BX-53, Shinjuku City, Tokyo, Japan) [15].

#### 2.3.5. Capacitation and Acrosome Reaction

The sperm capacitation and acrosome reaction status were assessed using chlortetracycline (CTC) fluorescent staining. To the 135 µL semen sample was added 15 µL H33258 solution (10 µg H33258/mL Phosphate Buffer Saline); then, it was incubated at room temperature for 10 min. Then, we added 250 µL of 2% (*w*/*v* in PBS) polyvinylpyrrolidone (Sigma-Aldrich, St. Louis, MO, USA) and centrifuged at 700× *g* for 5 min. The supernatant was discarded; then, we resuspended the pellet with 100 μL PBS and 100 μL CTC solution (750 mM CTC in 5 μL buffer consisting of 20 mM Tris, 130 mM NaCl, and 5 mM cysteine, pH 7.4). The sperm capacitation and acrosome reaction were assessed using a fluorescence microscope (Olympus BX53 Fluorescence Microscope, Shinjuku City, Tokyo, Japan) for 100 spermatozoa. The capacitated sperm were green fluorescence stained over the acrosomal region with a dark post-acrosomal region, while sperm not capacitated were green fluorescence stained evenly in the head. Sperm with a reacted acrosome were mottled green fluorescence stained over the head, green fluorescence only in the post-acrosomal region, or no fluorescence over the head [16].

### 2.4. Amino Acid Sequencing

The primer used for amplification and sequencing the subunit 1 NADH dehydrogenase (ND1) mtDNA was AB 736120.1 F: 5′ CCCATACCCTACCCCCTCAT 3′, R: 5′ GGGGTAGGATGCTCGGATTC 3′. The Sanger method was used for DNA sequencing. The results of DNA sequencing were presented in chromatograms and sequences (FASTA). DNA sequences were visualized in the chromatograms to evaluate the presence of mutations or variations by comparing the mutant chromatograms with normal sequences [14]. Alignment was performed using the ClustalW multiple sequence alignment algorithm in the MEGA 7 software, and the DNA sequences were translated into amino acid sequences using the same software. The analysis of amino acid variations in this study was carried out using the MEGA 7.0 software. The aligned 536 bp long DNA sequence was translated into amino acids, so that a 178 bp long amino acid sequence was obtained. Here, we also confirmed the modeling and amino acid sequence by downloading the NADH subunit 1 data displayed on Uniprot (www.uniprot.org (accessed on 3 January 2022)).

### 2.5. Data Analysis

Semen quality was randomly evaluated from six replicates in each group. The difference among treatments, equilibration period, and pre-freezing/post-thawed semen was analyzed using multivariate ANOVA and Pillai’s test. Sperm progressive motility is the primary indicator of post-thawed semen quality [6]. Therefore, regression analysis was used to identify the most potent pre-freezing parameter on post-thawed sperm motility.

The analysis was conducted based on the amino acids encoded by nicotinamide adenine dinucleotide hydride (NADH) dehydrogenase 1 (ND1) of mitochondrial deoxyribonucleic acid (mtDNA) alignment, which was followed by mapping the average frequency of mutation of amino acids in each group; the similarities and difference mutation of each amino acid; the amino acid mutation and their number mutant, mutation-based on individual amino acid, and amino acid mutations based on groups. Finally, the correlation coefficient of amino acid mutations to post-thawed Kacang buck semen parameters was calculated. All statistical analyses were conducted at a 95% significance level using Statistical Product and Service Solutions v.23 (IBM, New York, NY, USA).

## 3. Results

The obtained volume and concentration of fresh semen were 2.35 ± 0.21 mL and 3680.17 ± 165.96 sperm/mL, with sperm viability, sperm progressive motility, and IPM values of 91.50 ± 0.67, 88.17 ± 0.72, and 85.42 ± 0.79%, respectively (Table 1).

### 3.1. Effect of Equilibration Period and GTE Dose on Pre-Freezing and Post-Thawed Sperm Quality

All independent variables (GTE dose, equilibration period, pre-freezing, and post-thaw) had a significant effect (*p* < 0.05), either individually or together, on all dependent variables (sperm motility, sperm viability, IPM, MDA levels, and DNA fragmentation). The post-thawed semen quality improved with increased GTE doses up to a certain level. The T2 (0.1 mg GTE/100 mL extender) group provides the highest semen quality (*p* < 0.05) due to the highest sperm viability, sperm motility, and IPM, and the lowest MDA levels and DNA fragmentation. The T3 group (0.15 mg GTE/100 mL extender) had lower semen quality than those of the T2 group (*p* < 0.05). Semen quality was better when equilibrated in one hour than two hours. The sperm viability, sperm motility, and IPM of one-hour equilibration were significantly higher (*p* < 0.05) in the T2 group than those in two-hour equilibration periods in pre-freezing and post-thawed semen. Meanwhile, the MDA levels and DNA fragmentation of those were significantly lower (*p* < 0.05) (Table 2).

The T2 group showed the best quality of post-thawed semen based on the highest percentage of incapacitated sperm and the lowest of capacitated sperm and sperm with a reacted acrosome (*p* < 0.05). Those semen quality parameters in the T3 group were worse than those of the T2 group (*p* < 0.05). There were no significantly differences (*p* > 0.05) in those parameters between 1 h and 2 h equilibration in all dose groups (Table 3).

### 3.2. Regression Analysis for Identification of the Most Potent Pre-Freezing Parameter on Post-Thawed Sperm Motility

There were regression equations of all pre-freezing parameters, except the incapacitated sperm to the post-thawed motility. The coefficient of determination (R2) is in the range of 19.25–97.30% (*p* < 0.05), where the pre-freezing motility has the highest R2 to the post-thawed sperm motility (Table 4 and Figure 2). Therefore, frozen goat semen qualified for artificial insemination after reaching at least 40% post-thawed sperm motility. Based on the regression equation of Pt Mot = −8.07232 + 1.07062*Pf Mot (regression equation no.4, Table 4), pre-freezing sperm motility that met the minimum requirements was obtained at 44.90%. It means post-thawed sperm motility was 89.08% of pre-freezing sperm motility.

### 3.3. Mutation of Amino Acids Encoded by NADH Dehydrogenase 1 (ND1) of Mitochondrial Deoxyribonucleic Acid (mtDNA)

The amino acids encoded by NADH dehydrogenase 1 (ND1) of mitochondrial deoxyribonucleic acid (mtDNA) alignment can be seen in Figure 3. The random amino acid mutations in all groups seemly identified a pattern. Based on the average of mutation, the amino acids D and K were not mutated in all groups. The addition of 0.1 mg GTE per 100 mL SM–EY extender (T2) revealed the lowest frequency of amino acid mutation. Six amino acids were not mutated in T2, whereas those of other groups were mutated (Table 5). The mutations of amino acids individually based on their position revealed that 13 amino acids always had a mutation.

### 3.4. The Correlation Coefficient of Amino Acid Mutation and Post-Thawed Kacang Buck Semen Quality Parameter

Four amino acids D (aspartic acid), K (lysine), C (cysteine), and W (tryptophan) were not correlated (*p* > 0.05) to all post-thawed semen parameters. On the opposite, three of the amino acids, L (leucine), S (serine), and N (asparagine), were very strongly correlated (*p* < 0.05) to all post-thawed sperm parameters. The remaining 13 amino acids G (glycine), A (alanine), V (valine), I (isoleucine), T (threonine), E (glutamic acid), Q (glutamine), R (arginine), H (histidine), M (methionine), P (proline), F (phenylalanine), and Y (tyrosine) had varying correlations to the post-thawed sperm parameters (Table 6).

## 4. Discussion

Goat sperm are susceptible to freeze–thawing [17] due to the extreme temperatures and osmolarities change [18]. Cryoprotectants are needed by spermatozoa to survive during the freeze–thawing process. Ideally, cryoprotectants have a low molecular weight to penetrate cell membranes. Glycerol is an intracellular cryoprotectant, thereby preventing the ice crystals formation, which could lead to avoiding the rupture of the membrane [19].

### 4.1. GTE Dose

Without GTE addition, the post-thawed semen quality of the T0 group had the lowest sperm viability, motility, and IPM than the other groups. As mentioned earlier, the goat sperm is sensitive to oxidative stress. The cryopreservation process of semen leads to cold shock of sperm, which changes the ratio of polyunsaturated fatty acids (PUFA) and lowers the cholesterol content, causing instability of the sperm membrane. Plasma membrane integrity is essential for protecting the organelles of sperm and molecular transportation; thereby, it is crucial for sperm viability and sperm motility [20]. Sperms had antioxidant enzymes, such as glutathione peroxidase (GPX), catalase (CAT), and superoxide dismutase (SOD). However, the small volume of the sperm cytoplasm makes the transfer of these enzymes to the other part of the sperm challenging [21]. The concentration of antioxidant enzymes also decreased with the extender dilution. The unresolved oxidative stress causes a decrease in sperm motility and sperm viability [22]. Without GTE’s addition, the T0 group had the highest MDA level and DNA fragmentation of the groups. The high ROS generated freeze–thawing causes the peroxidation of double bonds of docosahexaenoic acid in PUFAs and produced MDA, which is a toxic lipid aldehyde species [23]. The high ROS may also lead to DNA fragmentation mediated by lipid peroxidation, modification of 8-hydroxy-guanine or 8-hydroxy-20-deoxyguanosine on mitochondrial and nuclear DNA base, destabilizing the DNA structure [24]. Sperm DNA is bound to protamine in a compact state, which helps protect the DNA against these destructive processes. Protamine forms disulfide bonds, making the sperm’s nucleus resistant to physical and chemical influences [25]. Protamine can condense and compact DNA to protect sperm from free radicals [26]. However, the interchain disulfide bridge opening on protamines during freeze–thawing may damage sperm DNA [27].

Post-thawed fertile sperm should have an intact plasma membrane to ensure the sperm viability, motility, and intactness of its DNA [28]. Treatment on semen of many species revealed varies results. Study on bull sperm indicated that the using of single-layer centrifugation selected spermatozoa resulted in less DNA damage [29]. Meanwhile, the addition of prostatic fluid to canine spermatozoa resulted in a higher proportion of spermatozoa with DNA damage after freezing–thawing [30]. Antioxidant administration can significantly decrease oxidative DNA damage [31]. GTE reserves the plasma membrane from lipid phosphorylation [32], including the acrosomal cap membrane necessary for the acrosomal reaction. During the freezing process, acrosome ruin and the partial displacement of the outer acrosomal membrane with the decimation of acrosomal enzyme content commonly occur due to ice crystal formation. This change causes swelling and rupture of the sub-acrosomal region. Changes in osmotic pressure will damage the lipid membrane structure, causing stress changes in proteins channel and ion leakage of the plasma membrane. This resulted in the cytoplasmic organs’ morphological changes and opened DNA to ROS attack [33]. The T1 group (with the addition of 0.05 mg GTE/100 mL extender) was of higher post-thawed semen quality than those of the CG. Green tea polyphenols may improve semen quality by reducing ROS production [34], lipid peroxidation, protein carbonylation, and DNA damage, thereby improving semen quality [35]. However, the number of these antioxidants is insufficient to offset the overproduction of ROS during freeze–thawing. The post-thawed semen quality of T1 was lower than that of T2.

The T2 group (added 0.1 mg GTE/100 mL extender) showed the highest post-thaw semen quality with the highest sperm viability, sperm motility, IPM, and lowest MDA levels and DNA fragmentation. This result is consistent with this finding that GTE’s addiction to an extender maintained the motility, viability, IPM, and DNA integrity of boar semen [36] and Simmental bull sperm [15]. On the same breed of goat (Kacang buck), the post-thawed semen quality of the T2 group was better than our ongoing study using AndroMed^®^ (minitube), namely 61.67 ± 0.29 vs. 46.19 ± 0.04% for sperm viability, 60.50 ± 0.26 vs. 40.83 ± 0.04% for sperm motility, and 58.58 ± 0.57 vs. 27.40 ± 0.04% for IPM, respectively. The EGCG, a bioactive substance of GTE, enhances post-thaw microscopic parameters and the fertility potential of buffalo spermatozoa [8]. The advantage effects of EGCG are due to the polyphenolic groups as antioxidant and chelating agents, whereas flavonoids act as catalysts on the membrane function [32]. Furthermore, the EGCG decreases oxidative stress parameters in spermatozoa [8]. Green tea polyphenols act on the adenosine monophosphate-activated protein kinase (AMPK), cyclic adenosine monophosphate (cAMP), calcium ions [34], ferrous iron, ferric iron, and low-density lipoprotein signaling pathways [35].

The T3 group (addition of 0.15 mg GTE/100 mL extender) showed a lower post-thawed semen quality than T2. Physiologically, sperm require low quantities of ROS for the healthy functioning of sperm [37]. The changes in ROS or antioxidant levels cause a redox imbalance. Excessive ROS or excessive antioxidants can disrupt the balance state [38]. High doses of GTE (T3 group) cause low levels of ROS and are inadequate for normal sperm function. Higher exposure to antioxidants leads to an antioxidant paradox that significantly reduces male fertility [31].

### 4.2. The Equilibration Period

The equilibration period is the duration of sperm contact with a cryoprotectant before freezing [39]. The equilibration stage affects the sperm’s ability to adapt to the extender environment to maintain homeostasis, osmotic tolerance, and cryo tolerance and prevent physical damage, osmotic stress, and cold stress during freezing [40]. Pre-freezing equilibration helps the sperm reach an osmotic equilibrium following the addition of the cryoprotectant. As mentioned earlier, the equilibration period of Kacang buck semen has not been established yet. Compared to other livestock species, the standard equilibration period for bull sperm before cryopreservation is 3–4 h to maintain sperm membrane integrity and motility [41]. In Jamunapari buck semen, a two or three-hour equilibration period resulted in the best post-thawed sperm [42]. Ahmad et al. reported that Beetal buck sperm survival is higher when equilibrated for 2–8 h [43].

The quality of post-thawed Kacang buck semen equilibrated for one hour was better than two hours for T1, indicating that extending the pre-freezing equilibration period did not improve the quality of post-thawed semen. However, this is consistent with another report that the prolonged pre-freezing equilibrated for 24 h of ram semen was not improved in its quality [44], and semen quality was improved neither in vitro nor in vivo [45]. Extending the equilibration period precisely reduces the quality of Kacang buck semen after thawing. Concerning antioxidants, when equilibrated for two hours, the sperm was excessively exposed to GTE’s antioxidants for more than the one-hour equilibration period, which caused a balance shift to reductive stress. Oxidative stress and reductive stress also harm sperm fertility [46]. Physiologically, sperm require low levels of ROS for normal functions such as sperm maturation, hyperactivation, acrosome reaction, and sperm–oocyte fusion [37]. ROS play an essential role in sperm fertility acquisition and tyrosine phosphorylation, sterol oxidation, and cholesterol outflow during the fertilization process [23]. Reductive stress is the powerful facet of redox imbalance that can harm sperm function. Decreased oxidative phosphorylation complex protein expression associated with mitochondrial dysfunction causes reductive stress [35].

A dose of 0.1 mg of GTE per 100 mL of skim milk–egg yolk (SM-EY) extender had the best effect on the quality parameters of Kacang buck semen. Changes in sperm viability, sperm motility, and IPM tendencies were reduced from pre-freezing to post-thaw and from 1 to 2 h equilibration. Meanwhile, the MDA concentrations and DNA fragmentation trends increased (Table 2). When equilibrated for one hour, the sperm motility of T2 was more than 40% (44.00% ± 0.19%), and its DNA fragmentation did not exceed 7% (6.43% ± 0.13%) (Table 2). This result was lower than that of Simmental bull sperm diluted in a 0.1 mg GTE per 100 mL SM–EY extender, which obtained 69.17% ± 1.4% motility and 3% ± 0.5% DNA fragmentation [15]. This difference indicated that Kacang buck sperm were more susceptible to oxidative stress than Simmental bull sperm. This result was also lower than the best quality of post-thawed Kacang buck semen diluted in an egg yolk–citrate extender supplemented with 2.5 mg of bull seminal plasma protein per milliliter, which showed 64.05% ± 0.45% progressive motility and 2.55% ± 0.75% DNA fragmentation [5]. These differences may be due to the antioxidant paradox [31] caused by GTE’s more potent antioxidant action than bull seminal plasma protein.

There were regression equations of all pre-freezing parameters, except the uncapacitated sperm to the post-thawed motility. The coefficient of determination (R2) is in the range of 19.25–97.30% (*p* < 0.05), where the pre-freezing motility has the highest R2 to the post-thawed sperm motility (Table 4 and Figure 1). Therefore, frozen goat semen that qualifies for artificial insemination should reach at least 40% post-thawed sperm motility. Based on the regression equation of Pt Mot = −8.07232 + 1.07062 × Pf Mot (regression equation no.4, Table 4), pre-freezing sperm motility that met the minimum requirements was obtained at 44.90%. It means post-thawed sperm motility was 89.08% of pre-freezing sperm motility.

### 4.3. Mutation of Amino Acids Encoded by NADH Dehydrogenase 1 (ND1) of Mitochondrial Deoxyribonucleic Acid (mtDNA)

This study is the first to report the amino acid mutations due to freeze–thawing. Based on semen quality parameters (sperm viability, motility, IPM, MDA levels, SDF, capacitation, and acrosome reaction), the T2 group had the highest quality. The frequency of amino acid mutations also revealed that T2 was the lowest. The results of this study were in accordance with our previous finding that the dose of 0.1 mg GTE/100 mL SM-EY extender is in the lowest nucleotide ND1 mtDNA mutation [14]. It was shown that the addition of 0.1 mg GTE per 100 mL SM–EY extender might protect the lipid bilayers and membrane function [32], decrease oxidative stress [8], reduce DNA fragmentation [35], and finally minimize the amino acid mutation.

Evolution theory revealed that maternal inheritance leads to the accumulation of mutations in mitochondrial DNA (mtDNA) that affect male fertility [47]. Mitochondria have multiple functions, including the synthesis of adenine triphosphate, production of reactive oxygen species, calcium signaling, thermogenesis, and apoptosis. Mitochondria significantly regulate the various physiological aspects of reproductive function from spermatogenesis to fertilization [48]. Each mitochondrion contains a covalently 16,569 bp DNA molecule that encodes 13 of the 83 subunits of the respiratory chain complex [49].

A study of mtDNA mutation related to sperm preservation was found less often. In human sperm, we reported the difference in the ND1 mtDNA of asthenozoospermia compared to normozoospermic in Egyptian men. The ND1 gene in the asthenozoospermia sample yields ten detected SNPs, six of which are synonymous mutations in nucleotides T3396C, T3423C, C3594T, G3693A, G3705A, A4104G, and non-synonymous mutation in T3398C, T3821C, G4048A and insertion (T) 4169. Conversely, normal sperm samples produced four SNPs: two synonymous mutations at nucleotides A4104G and A4158G and two non-synonymous mutations in T4216C and insertion (T) 4169 [50]. A higher prevalence of the mtDNA 4977 bp deletion was found in the subjects with impaired sperm motility and fertility. The mtDNA 4977 bp deletion, manganese superoxide dismutase (MnSOD, C47T), and 8-oxoguanine DNA glycosylase (hOGG1, C1245G) were accumulated in the sperm with poor motility [51]. Disruptions of mitochondrial DNA (mtDNA) may affect male reproductive function in the Iranian population. The 4977 and 7599 bp deletions of mtDNA may be genetic risk factors for male infertility [52]. Sixteen transition mutations have been predominantly detected withinside the Ghanaian populace’s mtDNA samples on the mutation of the amino acids. Missense mutations (threonine to alanine at positions 59, 112, 114, and 194 of the ATPase) present only in specific sperm abnormalities have been identified to contribute to male infertility [49].

A higher sperm mitochondrial DNA copy number (mtDNAcn) and deletion rate (mtDNAdel) were correlated with semen parameters and were associated with lower fertilization [53]. The mtDNA copy number was higher in asthenozoospermic semen samples and correlated negatively with sperm concentration, total sperm number, and motile spermatozoa. mtDNA content played a potential role as an indicator of semen quality. mtDNA copy number alterations and impaired chromatin integrity could affect reproductive success [54]. The mtDNA copy number amount was higher in aneuploid embryos than in those euploids. However, there were no statistically significant differences in mtDNA content related to embryo morphology, sex, maternal age, or implant ability [55].

### 4.4. The Correlation Coefficient of Amino Acid Mutation and Post-Thawed Kacang Buck Semen Quality Parameter

This study is the first to report the correlation of amino acid mutations to the semen quality parameters due to the freeze–thawing of animal semen. Three amino acid mutations (leucine, serine, and asparagine) strongly correlated with the post-thawed semen quality parameters. There was a significant correlation between a 7599 bp mtDNA deletion and asthenozoospermia in infertile Jordanian men. Furthermore, there was a significant correlation between 7599 bp mtDNA deletion and infertility, i.e., 63.6% in the infertile group compared to 34.2% in the fertile group. Additionally, the percentage of sperm motility showed a significant correlation to the frequency of that 7599 bp deletion [56]. There was a correlation between the Single-Nucleotide Polymorphisms (SNPs) and male infertility in a Saudi population [57].

Sperm fertilization capacities, including acrosin activity, acrosome reaction capability, and chromatin integrity, are related to mitochondrial functionality. Mitochondrial functionality might be necessary to maintain sperm acrosin activity, the acrosome reaction, and chromatin integrity [58]. The potential membrane of mitochondri is a pre-requisite for sperm motility, hyperactivation, capacitation, acrosin activity, the acrosome reaction, and DNA integrity. Therefore, the optimum mitochondrial activity is crucial for sperm function and semen quality. Defects of sperm mitochondrial function cause disturbance to the energy production required for sperm motility. Sperm mtDNA is sensitive to the oxidative damage and mutations that affect sperm function and lead to infertility. Abnormal semen parameters have a higher mtDNA copy number and reduced mtDNA integrity [48].

This study limited the extraction of green tea using ethanol solvent and the equilibration period to obtain semen quality that qualifies for artificial insemination. The one-hour equilibration period showed a satisfactory result. Meanwhile, the use of GTE antioxidants should be explored further. Among the different formulations, the nanoparticles are most effective in improving penetration into the cell [59] and have a higher surface area to volume ratio [60]. Future studies suggested using GTE nanoparticles on post-thawed semen quality, measuring the total antioxidant capacity and artificial insemination fields implementation. Since the post-thawed sperm motility is the crucial parameter of semen quality, attention needs to be focused on the glycine, valine, leucine, serine, and asparagine correlate strongly with post-thawed sperm motility. Sperm amino acid mutations due to SDF negatively impact pregnancy outcomes [61]. Our ongoing study reveals that SDF produces offspring with low fertility.

## 5. Conclusions

A 0.1 mg GTE per 100 mL skim milk–egg yolk extender and equilibrated in one hour resulted in the highest quality and minimum sperm amino acid mutation of Kacang buck semen. Interestingly, three amino acid mutations (leucine, serine, and asparagine) were strongly correlated with all post-thawed sperm parameter quality and post-thawed sperm motility in freeze–thawing. Future studies are needed to minimize the deterioration of semen quality, including SDF and sperm amino acid mutation, based on measuring the total antioxidant capacity.

## Figures and Tables

**Figure 1 vetsci-09-00403-f001:**
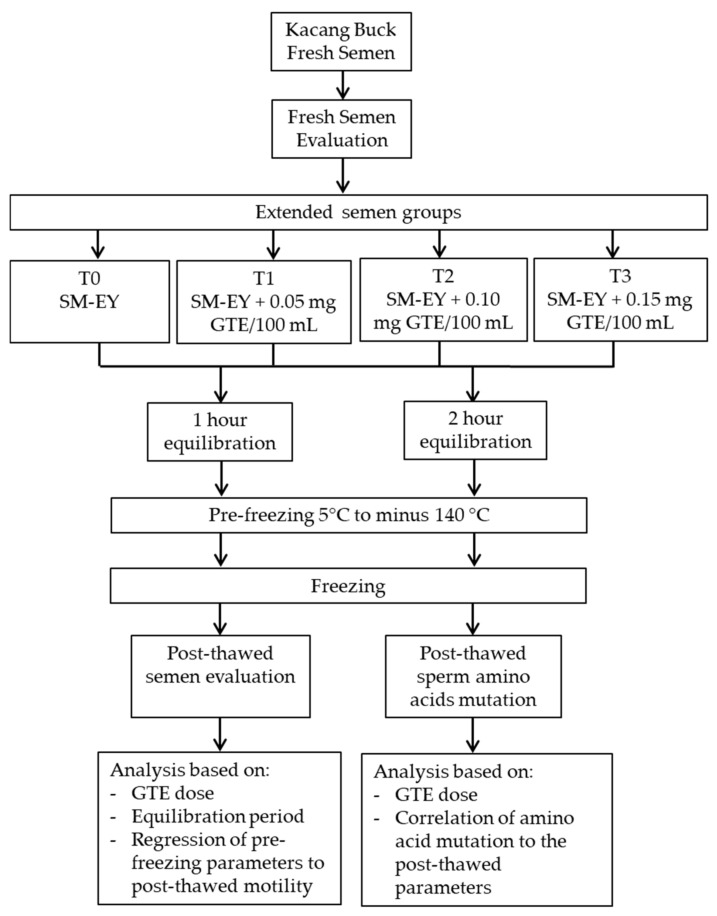
Work flow of the study of the green tea extract in the extender improved the post-thawed semen quality and decreased amino acid mutation of Kacang buck sperm.

**Figure 2 vetsci-09-00403-f002:**
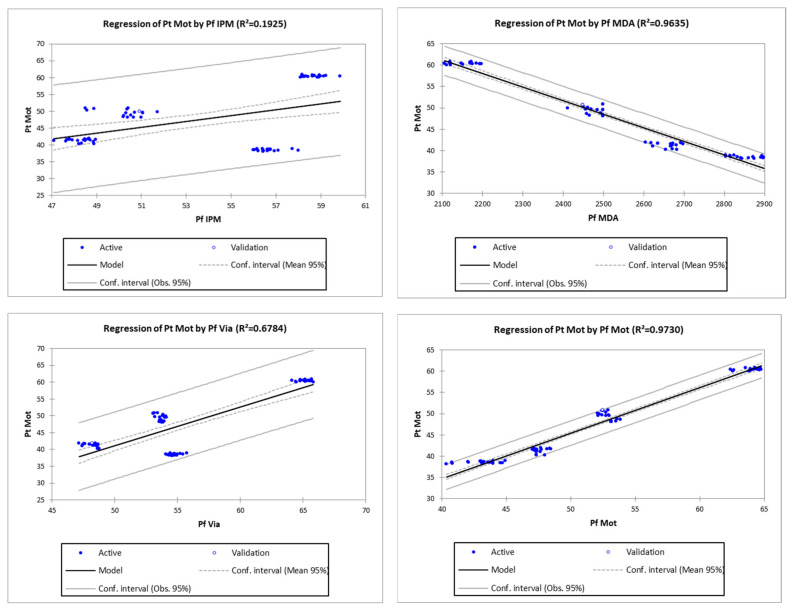
Linear regression curve of the post-thawed semen motility (Y) based on pre-freezing parameters (X).

**Figure 3 vetsci-09-00403-f003:**
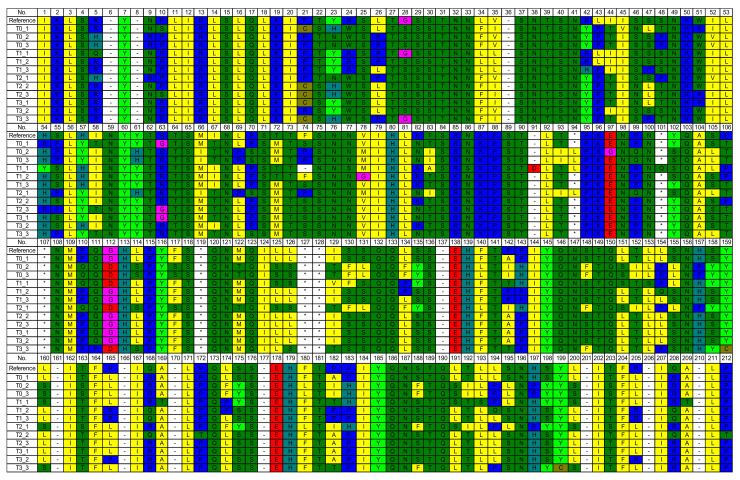
Amino acids encoded by nicotinamide adenine dinucleotide hydride (NADH) dehydrogenase 1 (ND1) of mitochondrial deoxyribonucleic acid (mtDNA) alignment.

**Table 1 vetsci-09-00403-t001:** The parameters of Kacang buck semen.

Parameters	Values
Volume (mL)	2.35 ± 0.21
Concentration (million/mL)	3680.17 ± 165.96
Viability (%)	91.50 ± 0.67
Progressive motility (%)	88.17 ± 0.72
Intact plasma membrane (%)	85.42 ± 0.79

**Table 2 vetsci-09-00403-t002:** Effect of green tea extract (GTE) in skim milk–egg yolk (SM-EY) extender on IPM, MDA levels, sperm viability, motility, and SDF of extended semen equilibrated for 1 or 2 h pre-freezing or post-thawed in Kacang buck sperm.

Parameters	Group	1 h Equilibration		2 h Equilibration	
		Pre-Freezing	Post-Thawed	Pre-Freezing	Post-Thawed
IPM	T0	56.68 ± 0.33 Ab	41.72 ± 0.38 Bc	38.50 ± 0.89 Cd	35.11 ± 0.41 Dc
(%)	T1	50.32 ± 0.94 Ac	46.52 ± 0.41 Bb	46.98 ± 0.51 Bb	41.39 ± 0.70 Cb
	**T2**	**58.73 ± 0.41 Aa**	**58.58 ± 0.57 Aa**	**53.77 ± 0.39 Ba**	**50.65 ± 0.55 Ca**
	T3	48.17 ± 0.32 Ad	46.60 ± 0.48 Ab	43.24 ± 0.88 Bc	39.99 ± 0.36 Cb
MDA	T0	2895.00 ± 183.72 Da	3258.20 ± 48.55 Ba	3018.60 ± 29.39 Ca	3378.48 ± 20.44 Aa
levels	T1	2498.76 ± 26.83 Dc	2858.76 ± 33.94 Bc	2618.76 ± 31.29 Cc	3102.37 ± 46.69 Ab
(nmol/mL)	**T2**	**2119.43 ± 7.96 Cd**	**2383.44 ± 26.29 Bd**	**2143.50 ± 40.16 Cd**	**2455.44 ± 16.97 Ac**
	T3	2671.58 ± 32.45 Cb	2983.58 ± 78.67 Bb	2683.58 ± 22.57 Cb	3342.42 ± 16.97 Aa
Sperm	T0	54.74 ± 0.44 Ab	39.20 ± 0.39 Cd	49.56 ± 0.36 Bb	35.07 ± 0.31 Dc
viability	T1	53.69 ± 0.37 Ab	50.40 ± 0.27 Bb	42.35 ± 0.28 Cc	38.15 ± 0.10 Db
(%)	**T2**	**65.10 ± 0.39 Aa**	**61.67 ± 0.29 Ba**	**57.47 ± 0.66 Ca**	**50.61 ± 0.44 Da**
	T3	48.23 ± 0.41 Ac	42.00 ± 0.42 Bc	41.09 ± 0.13 Bc	35.63 ± 0.43 Cc
Sperm	T0	42.97 ± 1.26 Ad	38.50 ± 0.19 Bd	37.33 ± 0.44 Bd	34.44 ± 0.26 Cc
motility	T1	52.73 ± 0.39 Ab	49.50 ± 1.23 Bb	41.59 ± 0.28 Cb	37.61 ± 0.60 Db
(%)	**T2**	**63.94 ± 0.72 Aa**	**60.50 ± 0.26 Ba**	**56.21 ± 0.43 Ca**	**49.71 ± 0.30 Da**
	T3	47.37 ± 0.39 Ac	41.25 ± 0.43 Bc	40.41 ± 0.50 Bc	34.99 ± 0.37 Cc
SDF	T0	4.05 ± 0.11 Ca	6.24 ± 0.10 Ba	4.19 ± 0.12 Ca	6.90 ± 0.15 Aa
(%)	T1	3.55 ± 0.19 Db	5.56 ± 0.08 Ab	4.22 ± 0.14 Ca	4.82 ± 0.09 Bc
	**T2**	**1.47 ± 0.11 Cc**	**4.31 ± 0.09 Ac**	**1.94 ± 0.05 Bc**	**4.32 ± 0.06 Ad**
	T3	3.42 ± 0.13 Bb	6.16 ± 0.05 Aa	3.62 ± 0.08 Bb	6.37 ± 0.10 Ab

Values with different superscripts A, B, C, D in the same row; a, b, c, and d in the same column of each parameters are significantly different (*p* < 0.05). SM-EY: extender without GTE; and T1, T2, and T3: SM-EY extender with the addition of 0.05, 0.1, and 0.15 mg GTE/100 mL extender, respectively. Values in bold were the best quality parameters of semen.

**Table 3 vetsci-09-00403-t003:** Effect of green tea extract (GTE) in skim milk–egg yolk (SM-EY) extender on the capacitation status and acrosome reaction of sperm of post-thawed Kacang buck equilibrated for 1 or 2 h.

Group	Incapacitated		Capacitated		Acrosome Reaction	
	1 h	2 h	1 h	2 h	1 h	2 h
T0	62.60 ± 0.55 Ad	60.20 ± 2.17 Ac	24.40 ± 1.34 Ba	26.40 ± 0.89 Ba	12.60 ± 1.34 Ca	13.40 ± 1.34 Ca
T1	69.40 ± 0.89 Ac	66.20 ± 2.68 Ab	21.00 ± 0.71 Bb	24.00 ± 2.24 Ba	9.60 ± 0.89 Ca	9.80 ± 0.45 Cb
**T2**	**80.60 ± 0.55 Aa**	**78.20 ± 1.79 Aa**	**13.00 ± 1.41 Bc**	**15.20 ± 1.30 Bc**	**6.40 ± 0.89 Cb**	**6.60 ± 0.55 Cc**
T3	74.20 ± 3.83 Ab	69.00 ± 4.80 Ab	20.40 ± 0.89 Bb	20.80 ± 0.45 Bb	11.40 ± 0.89 Ca	12.20 ± 1.30 Ca

Values with different superscripts A, B, C, D in the same row; a, b, c, and d in the same column of each parameters are significantly different (*p* < 0.05). SM-EY: extender without GTE; and T1, T2, and T3: SM-EY extender with the addition of 0.05, 0.1, and 0.15 mg GTE/100 mL extender, respectively. Values in bold were the best quality parameters of semen.

**Table 4 vetsci-09-00403-t004:** Regression equation of post-thawed motility (Pt Mot) based on pre-freezing semen quality parameters.

No	Parameters	Equation	*p*-Value	R	R^2^ (%)
1	Pf IPM	Pt Mot = 1.13776 + 0.86461 × Pf IPM	0.0001	0.44	19.25
2	Pf MDA	Pt Mot = 127.81664 − 0.03171 × Pf MDA	<0.0001	0.98	96.35
3	Pf Via	Pt Mot = −16.32235 + 1.15038 × Pf Via	<0.0001	0.82	67.84
**4**	**Pf Mot**	**Pt Mot = −8.07232 + 1.07062 × Pf Mot**	**<0.0001**	**0.99**	**97.30**
5	Pf SDF	Pt Mot = 70.34792 − 7.21669 × Pf SDF	<0.0001	0.88	77.62
6	Pf Acr	Pt Mot = 77.51510 − 3.43959 × Pf Acr	<0.0001	0.91	82.20
7	Pf Cap	Pt Mot = 82.35528 − 1.99723 × Pf Cap	<0.0001	0.90	81.48
8	Pf Unc	Pt Mot = 60.48636 − 0.15341 × Pf Unc	0.2797	0.13	1.69

Bold: highest correlation coefficient.

**Table 5 vetsci-09-00403-t005:** The average frequency of mutation of amino acids encoded by nicotinamide adenine dinucleotide hydride (NADH) dehydrogenase 1 (ND1) of mitochondrial deoxyribonucleic acid (mtDNA) in each group.

AA	Ref	T0		T1		T2		T3	
		n	%	n	%	n	%	n	%
G	2	2.00	100.00	1.00	50.00	0.50	25.00	1.67	83.33
A	3	1.00	33.33	1.33	44.44	0.00	0.00	1.00	33.33
V	2	2.00	100.00	1.33	66.67	1.00	50.00	2.00	100.00
L	28	5.60	20.00	4.67	16.67	2.00	7.14	5.00	17.86
I	17	3.00	17.65	1.00	5.88	0.00	0.00	0.67	3.92
S	30	9.00	30.00	4.00	13.33	1.75	5.83	5.67	18.89
T	19	6.00	31.58	4.67	24.56	1.75	9.21	3.33	17.54
D	0	0.00	0.00	0.00	0.00	0.00	0.00	0.00	0.00
E	3	0.20	6.67	0.00	0.00	0.00	0.00	0.00	0.00
N	19	2.00	10.53	2.00	10.53	0.50	2.63	2.33	12.28
Q	14	0.00	0.00	0.33	2.38	0.50	3.57	1.00	7.14
K	6	0.00	0.00	0.00	0.00	0.00	0.00	0.00	0.00
R	8	2.80	35.00	3.33	41.67	2.75	34.38	3.00	37.50
H	8	2.00	25.00	1.00	12.50	0.25	3.13	0.67	8.33
C	0	0.00	0.00	1.00	n/a	0.00	0.00	0.67	n/a
M	3	1.00	33.33	0.67	22.22	0.00	0.00	0.33	11.11
P	15	3.00	20.00	2.33	15.56	2.75	18.33	3.33	22.22
F	6	2.00	33.33	0.67	11.11	0.25	4.17	0.00	0.00
Y	10	1.00	10.00	2.33	23.33	0.50	5.00	2.00	20.00
W	1	0.00	0.00	1.00	100.00	0.00	0.00	0.00	0.00
Sum	194	42.60	21.96	32.67	16.84	14.50	7.47	32.67	16.84

Note: Ref: reference, the number of amino acid based on primer sequence, n: the average count of amino acid mutation. Red number: no mutation in all groups,  green highlighted number: no mutation in T2 group only.

**Table 6 vetsci-09-00403-t006:** Correlation coefficient of amino acid mutation and parameter of post-thawed Kacang buck semen.

Aa	Pt IPM	Pt MDA	Pt Via	Pt Mot	Pt SDF	Pt Inc	Pt Cap	Pt Acr
G	−0.98 *	0.95 *	−0.98 *	−0.98 *	0.94 *	−0.75 *	0.85 *	0.92 *
A	−0.68 *	0.77 *	−0.73 *	−0.72 *	0.68 *	−0.71 *	0.83 *	0.69 *
V	−0.93 *	0.9 *	−0.97 *	−0.97 *	0.93 *	−0.63 *	0.79 *	0.9 *
L	−0.94 *	0.97 *	−0.96 *	−0.96 *	0.91 *	−0.84 *	0.95 *	0.91 *
I	−0.86 *	0.86 *	−0.76 *	−0.75 *	0.71 *	−0.92 *	0.83 *	0.74 *
S	−0.98 *	0.96 *	−0.93 *	−0.93 *	0.88 *	−0.85 *	0.88 *	0.89 *
T	−0.85 *	0.9 *	−0.78 *	−0.77 *	0.72 *	−0.97 *	0.92 *	0.76 *
D	−0.10	0.13	−0.11	−0.11	0.07	−0.14	0.12	0.07
E	−0.74 *	0.7 *	−0.6 *	−0.6 *	0.56 *	−0.77 *	0.64 *	0.6 *
N	−0.80 *	0.85 *	−0.89 *	−0.88 *	0.85 *	−0.66 *	0.84 *	0.82 *
Q	−0.27 **	0.3 **	−0.09	−0.08	0.06	−0.63 *	0.37 *	0.13
K	−0.10	0.13	−0.11	−0.11	0.07	−0.14	0.12	0.07
R	−0.03	0.15	−0.11	−0.1	0.09	−0.2	0.31 **	0.1
H	−0.86 *	0.87 *	−0.76 *	−0.75 *	0.71 *	−0.95 *	0.86 *	0.74 *
C	−0.1	0.13	−0.11	−0.11	0.07	−0.14	0.12	0.07
M	−0.84 *	0.88 *	−0.76 *	−0.75 *	0.7 *	−0.97 *	0.91 *	0.75 *
P	−0.45 *	0.35 *	−0.53 *	−0.53 *	0.52 *	−0.02	0.17	0.46 *
F	−0.66 *	0.66 *	−0.5 *	−0.5 *	0.46 *	−0.83 *	0.65 *	0.52 *
Y	−0.27 **	0.37 *	−0.4 *	−0.39 *	0.38 *	−0.26 **	0.45 *	0.37 *
W	−0.14	0.01	−0.13	−0.14	0.15	− 0.20	0.18	0.1

Aa: amino acids symbol (G = glycine, A = alanine, V = valine, L = leucine, I = isoleucine, S = serine, T = threonine, D = aspartic acid, E = glutamic acid, N = asparagine, Q = glutamine, K = lysine, R = arginine, H = histidine, C = cysteine, M = methionine, P = proline, F = phenylalanine, Y = tyrosine, W = tryptophan). Pt: post-thawed, IPM: sperm intact plasma membrane, Via: sperm viability, Mot: sperm motility, SDF: sperm DNA fragmentation, Inc: incapacitated sperm, Cap: capacitated sperm, Acr: sperm with acrosome reaction. *: very significant correlation (*p* < 0.01), **: significant correlation (*p* < 0.05).

## Data Availability

The data presented in this study are available on request from the corresponding author.

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
