# Peer review of "Green Tea Extract in the Extender Improved the Post-Thawed Semen Quality and Decreased Amino Acid Mutation of Kacang Buck Sperm"

_vetsci, 2022, doi:10.3390/vetsci9080403_

Round 1

Reviewer 1 Report

In this paper the authors evaluated the effects of different doses of green tea extracts on Kacang buck semen that underwent the freeze/thaw procedure. They evaluated the reproductive potential of the semen analyzing some semen parameters such as mobility, progressive motility, capacitation and the state of the mitochondrial DNA in respect to some mutations in the AA. of the protein corresponding to the ND1 gene. They show that a one hour equilibration period in the extender containing 0.1mg/100 ml of GTE before the freezing procedure ensures the best semen quality after thawing. The paper is well organized in the experimental design even if it requires minor revisions that are detailed below. In any case I strongly suggest the publication in the following special issue: "New insights in Veterinary Theriogenology" of the section Veterinary Reproduction and Obstetrics.

REVISIONS REQUIRED:

Abstract

line 27: replace "qulity" with quality

line 30: ...strongly correlate...replace with ..strongly correlated to the post....

line 46: add a citation to support the concept.

Materials and methods

I think that all the "Materials and Methods" section needs to be integrated by a detailed explanation of the procedures instead of referring to previous published papers. 

lines 85-86: is not clear, please rewrite.

lines 106-107: please detail the procedure for DNA fragmentation assessement.

line 109: how did you evaluate progressive motility by CTC fluorescence? Sperm kinetic must be evaluated in objective way by means of a Computer Sperm Analyzer (CASA System)! can you detail the procedure?

Define all acronymous employed such as MDA, SDF,...

I think that to summarize all the procedure used to prepare samples a workflow diagram is necessary, please add it in this section.

lines 102-104: can you please precise the freezing procedure from 5°C to -140°C? At what distance from the bottom of the liquid nitrogen container was the straw suspended?

line 114: why did you amplify only a fragment of the gene? Are you sure that the remaining part of the corresponding protein has no mutation?

line 159: replace "sperm with acrosome reaction sperm" with"sperm with a reacted acrosome"

line 161: replace P<005 with P<0.05

line 161: replace "different" with differences

lines 161-166 use the same symbol in the text P or p to indicate the statistical significance!

line 188 replace "linier" with linear

line 283 eliminate a space between than/those

line 310 replace "cry" with cryo.

In my opinion the description of the AA alignement is difficult to understand and too long and this decreases the importance of the paper, please try to semplify the section.

In the new version the paper will suitable for publication.

Reviewer 2 Report

In this paper titled by “ Green tea extract in the extender improved the post-thawed semen quality and decreased amino acid mutation of Kacang buck sperm”, the authors observed the effects of GTE on frozen bull semen. This paper is understandable. However, the authors are suggested to further polish the language. I personally think this paper still need further corrections. The following listed comments are only represented my part comments. The authors should carefully revise this paper.

Comments:

1. in the abstract, the data related to sperm quality are not provided.

2. In this study, egg yolk and milk were both used as the protectants. Why? In addition, since there is harmful interaction between egg yolk and goat seminal plasma, the authors still used egg yolk. Why?

3. Please provide ethical statements about animal used.

4. I personally think the authors should briefly described the GTE preparation procedure.

5. when chilling and freezing, the contents are confusing. In addition, did they measure the chilling and freezing velocity?

6. please describe the procedure for sperm quality assessment using Chlortetracycline (CTC) fluorescence staining in detail.

7. please describe the procedure of Amino acid sequencing.

8. Most importantly, I am interested to know if they have checked the component in GTE? Which one is mainly for the cryoprotective effects?

9. in the results, what is SDF?

10. In the discussion, the authors repeatedly raised ROS. What is the effect of GTE on sperm ROS production?

Reviewer 3 Report

1   1. Should expand the terms ‘IPM’, MDA, and mtDNA at its first appearance in the manuscript.

2.     GTE preparation details should be provided. Was the extract, a cold extract or prepared using soxhlet apparatus?  

3.     Should provide the geographical indication of Green tea leaves collected from?

4.     Should provide animal ethical committee approval number and the reference to the guidelines followed during semen collection.

5.     Authors should include a positive control i.e., an effective commercially available extender in their study to compare their results.

6.     Authors should check viability; total and progressive sperm motility in the post-thawed samples with/without GTE infused extender.

7.     Analysis of mitochondrial membrane potential (an important sperm quality parameter) should be discussed.

8.     Graphs of figures 3 and 4 should be modified and also statistical analysis should be represented in the footnotes of the figures.

9.     Authors are suggested to include the following references at line 273, page No. 15 along with reference No. 28.

                                                    i.     Quality of bull spermatozoa after preparation by single-layer centrifugation. L Goodla, JM Morrell, Y Yusnizar, H Stålhammar, A Johannisson. Journal of dairy Science 97 (4), 2204-2212.

                                                  ii.     Effect of prostatic fluid on the quality of fresh and frozen-thawed canine epididymal spermatozoa. E Korochkina, A Johannisson, L Goodla, JM Morrell, E Axner. Theriogenology 82 (9), 1206-1211.

10. Authors should discuss their results with other researcher’s studies. Which way their study corroborates/contradicts with other researcher’s study?

11. Authors should check the conclusion part: Are the results really supporting the conclusion? The tested /selected parameters are sufficient for the conclusion?

12. References should be cited by following journal style/format.

13. Need to check for typographical errors, plagiarism, punctuation, and grammar throughout the manuscript.

1

Reviewer 4 Report

The manuscript is very interesting, but there are comments

In the materials and methods section, I would like to see a more detailed description of the assessment of sperm quality

Expand it MDA levels, IPM and SDF at the first mention

Results

In my opinion, a lot is devoted to amino acids. Too much information and forget what the manuscript is about. Please shorten it a little.

Discussion

Manuscript on the effect of green tea extract on sperm quality. I would like you to pay more attention to this

Round 2

Reviewer 2 Report

The authors have answered my concerns. It present style can be acceptable for publication.